



# A decision tree-based measure-correlate-predict approach for peak wind gust estimation from a global reanalysis dataset

Serkan Kartal[1], Sukanta Basu[1], and Simon J Watson[2]

[1]Faculty of Civil Engineering and Geosciences, Delft University of Technology, Delft, Netherlands
[2]Faculty of Aerospace Engineering, Delft University of Technology, Delft, Netherlands

**Correspondence:** Serkan Kartal (s.kartal@tudelft.nl)

**Abstract.** Peak wind gust ($W_p$) is a crucial meteorological variable for wind farm planning and operations. However, for many wind farm sites, there is a dearth of on-site measurements of $W_p$. In this paper, we propose a machine-learning approach (called INTRIGUE) that utilizes numerous inputs from a public-domain reanalysis dataset, and in turn, generates long-term, site-specific $W_p$ series. Through a systematic feature importance study, we also identify the most relevant meteorological variables for $W_p$ estimation. Even though the proposed INTRIGUE approach performs very well for nominal conditions compared to specific baselines, its performance for extreme conditions is less than satisfactory.

## 1 Introduction

Wind gust or gusty wind is a common household term. However, there has yet to be a consensus on its exact scientific definition. For example, according to the Glossary of Meteorology (AMS, 2023), a wind gust can be defined as:

"A sudden, brief increase in the speed of the wind. It is of a more transient character than a squall and is followed by a lull or slackening in the wind speed. [...] According to U.S. weather observing practice, gusts are reported when the peak wind speed reaches at least 16 knots and the variation in wind speed between the peaks and lulls is at least 9 knots. The duration of a gust is usually less than 20 s."

A somewhat different definition has been suggested by the U.S. National Oceanic and Atmospheric Administration (NOAA, 2023):

"A rapid fluctuation of wind speed with variation of 10 knots or more between peaks and lulls."

As opposed to these quantitative definitions, the World Meteorological Organization (WMO, 2021, page 227) describes wind gusts in a very generic way:

"The extent to which wind is characterized by rapid fluctuations is referred to as gustiness, and single fluctuations are called gusts."

Despite these vast differences in the definition of wind gusts, most sources seem to agree on the meaning of 'peak' wind gusts ($W_p$):





"The maximum observed wind speed over a specific time interval." (WMO, 2021, page 227)

On the basis of this definition, it is appropriate to assert that "peak gust need not be a true gust of wind" (AMS, 2023). For
quiescent atmospheric settings, within certain time periods, peak wind gusts may very well be close to near-calm conditions.
While in the presence of certain meteorological phenomena (e.g., downbursts, tornadoes), they may attain severe, hazardous
intensities. The focus of the current study is on the estimation of a wide range of peak wind gusts using a decision tree (DT)-
based machine learning (ML) approach.

Measurements of $W_p$ require high-frequency observations. Typically, cup, propeller, and sonic anemometers record wind
speeds with sampling rates of O(1–10) Hz. First, block averaging is performed on these measured time series with a window
length of $\tau$ seconds. Subsequently, for a specific time period $T$, the maximum (or peak) of the $\tau$-sec-averaged values is esti-
mated, which is known as the $\tau$-sec peak wind gust (Panofsky and Dutton, 1984; Holmes, 2001; Solari, 2019). The magnitude
of $W_p$ strongly depends on the selected values of $\tau$ and $T$ (Brook and Spillane, 1968; Beljaars, 1987). Most commonly, $\tau$ is
chosen to be equal to a few seconds. Depending on the application, the value of $T$ can be as small as a few minutes to sev-
eral hours. For example, the Automated Service Operations System (ASOS) employed by the U.S. National Weather Service
measures 5-sec peak wind gusts and considers a time period of one minute. In contrast, in the wind energy literature (Rohatgi
and Nelson, 1994), the combination of $\tau = 3$-sec and $T = 10$-min are more prevalent. From a wind engineering perspective, a
historical account of 3-sec $W_p$ has recently been documented by Lombardo (2021).

If the mean and peak gust wind speeds during $T$ are denoted by $W$ and $W_p$, respectively, then one can write (Holmes, 2001):

$$W_p = W + c\sigma_W. \tag{1}$$

Here $\sigma_W$ is the standard deviation of wind speed. If the high-frequency wind speed data follows a Gaussian distribution during
$T$, then $c$ can be approximately equal to 3.5 ($\approx$99.98 percentile). Eq. (1) can be re-written as:

$$W_p = W \left(1 + c\frac{\sigma_W}{W}\right), \tag{2a}$$

or,

$$G = \frac{W_p}{W} = \left(1 + c\frac{\sigma_W}{W}\right). \tag{2b}$$

The ratio $G$ is the so-called gust factor. Whereas the ratio $\left(\frac{\sigma_W}{W}\right)$ is known as the turbulence intensity (TI).

In the wind energy literature, several studies (Sumner and Masson, 2006; Wharton and Lundquist, 2012; Hedevang, 2014;
Siddiqui et al., 2015; St Martin et al., 2016; Lee et al., 2020) have reported on the (negative) impacts of high TI on power
production. Given the linear relationship between $G$ and TI, it is expected that high value gust factors may also be responsible
for sub-optimal wind power production. Highly fluctuating power production due to wind gusts may also cause problems for
electrical grid balancing (Milan et al., 2013). In addition to power production, high TI (or $G$) also induces significant fatigue
loading on wind turbines (Kelley et al., 2000; Hansen and Larsen, 2005; Dimitrov et al., 2017; Ebrahimi and Sekandari, 2018;
Ren et al., 2018; Asadi and Pourhossein, 2021).





Contemporary wind turbine design standards (e.g., IEC, 2019) include provisions for extreme weather conditions. Some of them are related to extreme wind gusts (e.g., extreme coherent gust with direction change, extreme operating gust). Severe meteorological phenomena, such as thunderstorm downbursts, tornadoes, and hurricanes, can generate extreme wind gusts. We document a few historical events of relevance. One of the highest-ever recorded gusts was recorded at the Andrews Air Force Base on August 1, 1983 (Fujita, 1985). Due to the passage of a microburst, near-surface gust speed reached approximately 130

knots ($\approx 67$ m s$^{-1}$). The airplane of U.S. President Ronald Reagan landed just six minutes earlier than this extreme gust event. This extreme event provided the necessary stimulus to mobilize extensive research on microburst phenomena in the 1980s. Petersen et al. (1998) documented an even stronger gust event in a review paper on wind power meteorology. They analyzed wind data during a storm event on the Faroe Islands. There, prior to collapsing, one of the instrumented met-towers registered a gust value of 76.7 m s$^{-1}$. It is entirely possible that these types of severe gust events might hamper the structural integrity of

modern-day wind turbines. About twenty years ago, one such event took place on Miyakojima island in Japan. The recorded maximum gust speed was 74.1 m s$^{-1}$. Out of six turbines, three turbines entirely collapsed, and the other ones sustained significant damage (Ishihara et al., 2005). A more recent event was documented by Hawbecker et al. (2017). A thunderstorm producing multiple downbursts and tornadoes passed through the Buffalo Ridge Wind Farm, Minnesota (USA) in 2011. The resulting wind gusts caused damage to turbine blades and also caused buckling of a turbine tower.

Based on the aforementioned published studies and other anecdotal evidence, we can conclude that both nominal and extreme wind gusts are critical for wind energy. Therefore, during the wind farm planning and operation stages, the (detrimental) effects of wind gusts should be adequately accounted for. However, it is widely known in the literature that wind gusts are spatially and temporally highly intermittent. Thus, the long-term statistical characterization of such events utilizing on-site wind sensors is rather challenging and expensive. As an alternative, mesoscale meteorological models (MMMs) can be used to predict and

forecast peak wind gusts (Goyette et al., 2003; Ágústsson and Ólafsson, 2009; Stucki et al., 2016; Kurbatova et al., 2018). Typically, different physical parameterizations are used for convective and non-convective gusts (refer to Sheridan, 2011, and the references therein). Although these physical parameterizations have improved over the years, considerable improvements can still be made. It is also important to note that MMMs are computationally expensive, especially when sub-kilometer grids and gray-zone physical parameterizations (Boutle et al., 2014; Shin and Hong, 2015) are used. In this paper, we propose a data-

based alternative approach that leverages a decision tree-based technique for peak wind gust estimation from a global reanalysis dataset. We name the proposed approach: INTRIGUE (decIsioN TRee-based wInd GUst Estimation). It requires limited (say one year) on-site $W_p$ data for training and can generate a multi-year $W_p$ time-series for that specific site. It also performs reasonably well for generating $W_p$ data for neighboring sites. Most importantly, separate parameterizations for convective and non-convective events are not required.

The structure of this paper is as follows. Since the proposed INTRIGUE approach uses various meteorological input features (e.g., friction velocity, CAPE), we briefly summarize a few relevant physical parameterizations in Section 2. In Section 3, we include a concise literature review on various applications of ML in wind gust-related research. Descriptions of the study area and relevant datasets are provided in Sections 4 and 5, respectively. Various technical details pertaining to the INTRIGUE approach (e.g., data splitting, hyperparameter turning) are elaborated in Section 6. In Section 7, we report all the results



including a discussion on feature importance. The limitations of the INTRIGUE approach for extreme wind gusts are mentioned in Section 8. Concluding remarks and future perspectives are provided in Section 9.

## 2   Physical Parameterizations of Peak Wind Gusts

In a technical report, Sheridan (2011) provided a comprehensive review of various physical parameterizations for peak wind gusts. A few years later, Kurbatova et al. (2018) investigated the capabilities of seven of these parameterizations in forecasting
gusts in Russia. Here, we briefly mention a few well-known (and simple) parameterizations. Unless stated explicitly, we assume $W$ and $W_p$ are defined at the height of 10 m above ground level.

It is well-known in the literature that the gust factor ($G$) depends on $\tau$, $T$, measurement height, wind direction, surface roughness, and other factors (Wieringa, 1973; Ashcroft, 1994; Weggel, 1999; Choi and Hidayat, 2002; Harris and Kahl, 2017). However, for simplicity, in constant gust factor parameterization, $G$ is assumed to be equal to a constant $c_{GF}$:

$$G = \frac{W_p}{W} = c_{GF}. \tag{3}$$

A few climatological studies have found that even though G varies significantly with respect to underlying topography, the spatially averaged value of G is quasi-universal. For example, Harris and Kahl (2017) analyzed multi-year, high-resolution ASOS data from Milwaukee (USA) and reported an average value of $c_{GF} = 1.74$. While analyzing Santa Ana winds in Southern California (USA), Fovell and Cao (2017) found $c_{GF}$ = 1.6-1.7 to be representative for two locations. Based on multi-year
observational data from more than thirty stations in Switzerland, Stucki et al. (2016) estimated $c_{GF}$ to be equal to 1.67.

The following surface layer similarity-based formulation is also often used for non-convective conditions (Sheridan, 2011; Stucki et al., 2016):

$$W_p = W + c_{u_*} u_*. \tag{4}$$

Here $u_*$ is the so-called surface friction velocity. The coefficient $c_{u_*}$ is on the order of 7.5. Sometimes, in Eq. 4, a nonlinear
function of the stability parameter is used in conjunction with the $c_{u_*} u_*$ term (e.g., ECMWF, 2020).

Certain non-convective formulations make use of boundary layer height ($H$, in m) and/or wind speed at boundary layer height ($W_H$) in a semi-empirical manner. Stucki et al. (2016) reported one such formulations:

$$W_p = W + (W_H - W)\left(1 - \frac{H}{2000}\right). \tag{5}$$

Brasseur (2001) proposed an interesting physically-based approach for gust estimation. It assumes that the gusts at the
surface originate from the upper part of the boundary layer. Since the formulation is somewhat involved, we do not include it here. However, we do point out that it includes vertically-averaged turbulent kinetic energy ($\overline{e}$) as a key variable.

In the proposed INTRIGUE approach, we use $W$, $u_*$, $H$, and several other relevant meteorological variables (e.g., surface sensible heat flux, CAPE). If a relevant variable is not available as an input feature, we use our domain knowledge to include a surrogate variable. For example, $\overline{e}$ is not available in the global reanalysis dataset that we used. Hence, as a substitute, we



**Figure 1.** Top left panel: digital elevation map of the study area. The symbols denote the locations of three West Texas Mesonet stations. Photographs of the stations at the REESE Technology Center (Lubbock county), MACY (Garza county), and FLUVANNA (Borden county) are shown at top-right, bottom-left, and bottom-right panels, respectively. These photographs are downloaded from: https://www.mesonet. ttu.edu/

.

make use of the average energy dissipation rate ($\overline{\varepsilon}$) in the boundary layer. The relationship between $\overline{e}$ and $\varepsilon$ has been studied in the literature (e.g., Basu et al., 2021). In Section 7 of this paper, we perform a systematic feature importance study and show that most of the variables included in well-known physical parameterizations (e.g., Eqs. 3–5) also turn out to be very important from a purely data-based ML standpoint.





## 3 Applications of ML in Wind Gust Research

To the best of our knowledge, only a handful of studies (Mercer et al., 2008; Sallis et al., 2011; Chaudhuri and Middey, 2011; Carcangiu et al., 2014; Patlakas et al., 2017; Wang et al., 2020; Spassiani and Mason, 2021; Schulz and Lerch, 2022; Wang et al., 2022) have incorporated machine-learning approaches for wind gust-related research. Several of these studies focused on extreme wind gusts. For example, Mercer et al. (2008) studied downslope windstorms in Colorado (USA). They compared the performance of stepwise linear regression, support vector regression, and multilayer perceptrons in short-term forecasting of

extreme wind gusts. They utilized various meteorological variables (e.g., 700 hPa wind speed, mountaintop relative humidity) and parameters derived from radiosondes (e.g., integrated Scorer parameter, Sangster parameter) as input features. In another study, Chaudhuri and Middey (2011) used ML approaches for predicting peak wind gusts associated with pre-monsoon thunderstorms near Kolkata (India). Their newly developed adaptive neuro-fuzzy interference system outperformed multiple linear regression, radial basis function network, and multilayer perceptrons.

Various ML approaches (e.g., Kalman filtering, Gaussian Process regression) were also utilized for short-term forecasting of wind gusts. Some of these studies post-processed numerical weather prediction data (e.g., Patlakas et al., 2017; Schulz and Lerch, 2022; Wang et al., 2022). In contrast, Wang et al. (2020) only used observed time-series data from Jiangsu province (China) for forecasting. They used an ensemble learning method comprising of Random Forest, Long Short-Term Memory, and Gaussian Process regression. In order to mitigate wind turbine loads, Carcangiu et al. (2014) proposed a multilayer perceptron

for gust detection followed by an innovative turbine control strategy.

Numerous studies (e.g., Enloe et al., 2004; Azorin-Molina et al., 2016; Brázdil et al., 2017; Lombardo and Zickar, 2019) have reported climatologies and in-depth statistical analysis of wind gusts in various countries. However, they do not leverage any ML approaches. An exception is the study by Spassiani and Mason (2021). They used Self-Organizing Maps (Kohonen, 1990, 2013) to perform automated classification of wind gusts in Australia in order to identify their dynamical origins.

It is important to stress that the scope of the present study is different from these past ML-based investigations. We are interested in generating long-term, site-specific peak wind gust ($W_p$) series based on a global reanalysis dataset. Our proposed INTRIGUE approach, described in Section 7, can be described as an advanced Measure-Correlate-Predict (MCP) approach for peak wind gusts. MCP is well-established in wind resource estimation (e.g., Rogers et al., 2005; Carta et al., 2013). However, its usage in peak wind gust estimation is not known to us.

## 150 4 Study Area

This study focuses on the West Texas Panhandle region, one of the largest semi-arid regions in the world. This region's major distinguishing topographical feature is the Caprock Escarpment (see top-left panel of Fig. 1), a precipitous cliff with an average height of ∼90 m. Otherwise, this region is very flat, homogeneous, and sparsely vegetated. Owing to the frequent occurrence of strong nocturnal low-level jets, the wind resource of this region is very good (wind class 3-5). This fact has led to the

construction of numerous wind farms in this region, some of which (e.g., Roscoe, Horse Hollow, Buffalo Gap, Sweetwater) are among the largest operating wind farms in the U.S.



**Table 1.** A partial list of ERA5 and derived variables utilized as input features for the INTRIGUE approach

| Type | Variable | Equation | Description | Units |
|---|---|---|---|---|
| Raw | $W_{p10}^i$ | | Instantaneous wind gust at 10 m AGL (called $i10fg$ in ERA5) | m s$^{-1}$ |
| Raw | $W_{p10}^m$ | | Mean wind gust at 10 m AGL since previous post-processing (called $10fg$ in ERA5) | m s$^{-1}$ |
| Derived | $W_{10}$ | $\sqrt{U_{10}^2 + V_{10}^2}$ | Wind speed at 10 m AGL computed from zonal and meridional components | m s$^{-1}$ |
| Derived | $W_{100}$ | $\sqrt{U_{100}^2 + V_{100}^2}$ | Wind speed at 100 m AGL computed from zonal and meridional components | m s$^{-1}$ |
| Derived | $\alpha$ | $\log\left(W_{100}/W_{10}\right)/\log(100/10)$ | Power-law exponent of wind profile within 10–100 m AGL | – |
| Derived | $\beta$ | | Change in wind direction between 10 m and 100 m AGL | degrees |
| Raw | $T_2$ | | Air temperature at 2 m AGL (called $t2m$ in ERA5) | K |
| Raw | $T_0$ | | Skin temperature (called $skt$ in ERA5) | K |
| Raw | $T_s$ | | Upper-level soil temperature (called $stl1$ in ERA5) | K |
| Raw | $T_{d2}$ | | Dewpoint temperature at 2 m AGL ($d2m$) | K |
| Derived | $\Delta T_1$ | $T_2 - T_0$ | Difference of air and skin temperatures | K |
| Derived | $\Delta T_2$ | $T_0 - T_s$ | Difference of skin and soil temperatures | K |
| Derived | $\Delta T_3$ | $T_2 - T_{d2}$ | Temperature dew point spread | K |
| Raw | $u_*$ | | Surface friction velocity (called $zust$ in ERA5) | m s$^{-1}$ |
| Raw | $\tau_{ew}$ | | Instantaneous $X$ surface stress (called $iews$ in ERA5) | N m$^{-2}$ |
| Raw | $\tau_{ns}$ | | Instantaneous $Y$ surface stress (called $inss$ in ERA5) | N m$^{-2}$ |
| Raw | $\bar{\varepsilon}$ | | Energy dissipation rate in boundary layer (called $bld$ in ERA5) | J m$^{-2}$ |
| Raw | $\bar{\varepsilon}_m$ | | Mean energy dissipation rate in boundary layer (called $mbld$ in ERA5) | W m$^{-2}$ |
| Raw | $H_S$ | | Instantaneous surface sensible heat flux (called $ishf$ in ERA5) | W m$^{-2}$ |
| Raw | $H_L$ | | Instantaneous moisture flux (called $ie$ in ERA5) | Kg m$^{-2}$ s$^{-1}$ |
| Raw | $H$ | | Boundary layer height (called $blh$ in ERA5) | m |
| Raw | $P_0$ | | Mean sea level pressure (called $msl$ in ERA5) | Pa |
| Raw | TCC | | Total cloud cover (called $tcc$ in ERA5) | – |
| Raw | LCC | | Low-level cloud cover (called $lcc$ in ERA5) | – |
| Raw | CAPE | | Convective available potential energy (called $cape$ in ERA5) | J kg$^{-1}$ |
| Raw | CIN | | Convective inhibition (called $cin$ in ERA5) | J kg$^{-1}$ |
| Derived | HRSin | $\sin(2\pi\text{Hour}/24)$ | Sine-encoding of hours | – |
| Derived | HRCos | $\cos(2\pi\text{Hour}/24)$ | Cosine-encoding of hours | – |
| Derived | DYSin | $\sin(2\pi\text{Day}/365)$ | Sine-encoding of Julian days | – |
| Derived | DYCos | $\cos(2\pi\text{Day}/365)$ | Cosine-encoding of Julian days | – |
| Derived | MOSin | $\sin(2\pi\text{Month}/12)$ | Sine-encoding of months | – |
| Derived | MOCos | $\cos(2\pi\text{Month}/12)$ | Cosine-encoding of months | – |

The West Texas Mesonet (henceforth WTM) is a high-density network of automated surface meteorological stations which spans the West Texas Panhandle region and extends to some parts of New Mexico and Colorado. This network (www.mesonet. ttu.edu) was established in 1999 by the Atmospheric Science Group at Texas Tech University (Schroeder et al., 2005).

For the purpose of this study, we have selected three WTM stations (called REESE, MACY, and FLUVANNA) which are located in areas of varying topographical complexities. Their locations are demarcated by various symbols in the digital





elevation map of Figure 1. The station at the REESE Technology Center is located at latitude $33°\ 36'\ 26''$ N, longitude $102°\ 02'\ 55''$ W, and elevation 1021 m, about 19 km west of the city of Lubbock, Texas. The topography is very flat surrounding this station (see the photograph in the top-right panel of Figure 1). The MACY station is located at the edge of the Caprock

Escarpment (bottom-left panel of Figure 1). Given the complex topographical surroundings, more gusty wind conditions are prevalent at this site. The latitude, longitude, and elevation of this station are: $33°\ 4'\ 53''$ N, $101°\ 30'\ 58''$ W, 874 m, respectively. The FLUVANNA station is situated on a relatively flat area off the Caprock (refer to the bottom-right panel of Figure 1). However, a few kilometers away from the station, the ruggedness of the topography increases substantially. This station is located at latitude $32°\ 53'\ 57''$ N, longitude $101°\ 12'\ 7''$ W, and at an elevation of 826 m, about 105 km south-east of Lubbock,

Texas.

## 5 Description of Observed and Reanalysis Datasets

Each station in the WTM network measures a multitude of meteorological variables. However, in this study, we only utilize the 3-sec peak wind gust ($W_p$) data from the REESE, MACY, and FLUVANNA stations. The associated anemometers (R. M. Young propeller type) are located at 10 m above ground level. Technical details about the measuring instruments, data quality

control, sensor calibration, and other aspects can be found in Schroeder et al. (2005).

In conjunction with these observed $W_p$ data, we make use of several meteorological variables (including simulated wind gusts) from a global reanalysis dataset known as ERA5 (Hersbach et al., 2020). ERA5 is the fifth-generation reanalysis product of the European Centre for Medium-Range Weather Forecasts. The horizontal resolution of this reanalysis dataset is approximately 32 km. For each of the three WTM stations (i.e., REESE, MACY, and FLUVANNA), we have extracted ERA5 data

from the corresponding nearest grid points. In Table 1, we list some of the extracted ERA5 variables as well as a few derived ones. In total, 265 input features are used in the INTRIGUE approach.

In the ERA5 dataset, snapshots of most of the meteorological variables are output every hour. Whereas, in the case of the WTM, the variables are temporally averaged with a sampling rate of 5 minutes. Direct comparison of point measurements against atmospheric model-generated gridded-data is an ill-posed problem. We do not attempt to resolve this issue in this paper.

However, to avoid the sampling rate mismatch between the WTM and the ERA5 datasets, we preprocess the WTM data with a moving-maxima filter with a non-overlapping window of one hour. For example, we compute the maximum of contiguous 12 $W_p$ samples measured during 13:30–14:30 CST to estimate the corresponding 'hourly' value of $W_p$ at 14:00 CST.

In Figure 2, we plot several bi-variate histograms. In the $x$-axes, we have the predictor variables – i.e., the meteorological variables from the ERA5 dataset. In $y$-axes, the peak wind gusts (i.e., $W_p$) from the WTM stations are shown as predictands.

It is evident that both instantaneous wind gusts ($W_{p10}^i$) and friction velocity ($u_*$) from ERA5 are strongly correlated with the measured $W_p$ data ($r^2$ is on the order of 0.8). In contrast, the correlations between boundary layer heights ($H$) from ERA5 and $W_p$ values are much weaker ($r^2 \approx 0.5$). The proposed INTRIGUE approach, described in Section 7, exploits not only the strong correlations but also the weaker ones in a systematic manner to provide a more accurate prediction of $W_p$.



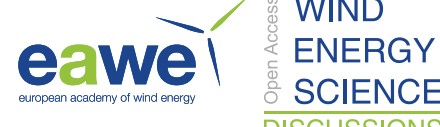

**Figure 2.** Bi-variate histograms of several meteorological variables. In the $x$ axis, the predictor variables from the ERA5 dataset are plotted. The predictor variables are: $W_{p10}^i$ (left panels), $u_*$ (middle panels), and $H$ (right panels). The predictand variable, $W_p$, is plotted in the $y$ axis. The top, middle, and bottom panels correspond to the REESE, MACY, and FLUVANNA stations, respectively. To enhance the clarity of these plots, we do not show the data points where $W_p > 25$ m s$^{-1}$. In the bottom-right corner of each plot, we report the Pearson's correlation coefficient ($r$).



**Table 2.** Hyperparameter search spaces of the bagging and boosting ML models. For each WTM station, different optimized models are constructed. In the last three columns, the best configurations of the models are reported when data from the year 2003 are used for training.

| Algorithm | Hyperparameter | Range | REESE | MACY | FLUVANNA |
|---|---|---|---|---|---|
| Random Forest (RF) | tree num | [4, min(2048, # instance)]] | 276 | 827 | 45 |
| | max_features | [0.1, 1] | 0.260 | 0.280 | 0.730 |
| | leaf num | [4, 32768] | 3454 | 3259 | 321 |
| Extremely Randomized Trees (ERT) | tree num | [4, min(2048, # instance)]] | 28 | 146 | 25 |
| | max_features | [0.1, 1] | 0.780 | 0.330 | 0.990 |
| | leaf num | [4, 32768] | 597 | 2721 | 3454 |
| Extreme Gradient Boosting (XGB) | tree num | [4, min(32768, # instance)]] | 393 | 712 | 73 |
| | leaf num | [4, min(32768, # instance)]] | 44 | 17 | 480 |
| | min child weight | [0.001, 128] | 5.540 | 0.009 | 54.000 |
| | learning rate | [0.001, 0.1] | 0.022 | 0.017 | 0.084 |
| | subsample | [0.1, 1.0] | 0.950 | 0.540 | 1.000 |
| | reg alpha | [0.001, 1024] | 0.410 | 0.001 | 0.400 |
| | reg lambda | [0.001, 1024] | 16.430 | 11.220 | 0.001 |
| | colsample by level | [0.01, 1.0] | 0.840 | 0.270 | 0.210 |
| | colsample by tree | [0.01, 1.0] | 0.850 | 0.800 | 0.640 |
| Light Gradient Boosting Machine (LGBM) | tree num | [4, min(32768, # instance)]] | 6909 | 473 | 439 |
| | leaf num | [4, min(32768, # instance)]] | 24 | 29 | 162 |
| | min child samples | [2, 129] | 2.000 | 13.000 | 3.000 |
| | learning rate | [0.001, 0.1] | 0.002 | 0.017 | 0.020 |
| | reg alpha | [0.001, 1024] | 3.350 | 0.001 | 0.001 |
| | reg lambda | [0.001, 1024] | 0.002 | 0.096 | 0.011 |
| | max bin | [3, 11] | 5 | 7 | 6 |
| | colsample by tree | [0.01, 1.0] | 0.610 | 0.920 | 0.510 |

## 6 Proposed INTRIGUE Approach

In the following sub-sections, we describe various technical details associated with the proposed INTRIGUE approach.

### 6.1 Strategy for Splitting of Available Data

In this study, we have eleven years (2003 to 2013) of WTM and ERA5 datasets at our disposal. Instead of training various ML models with lots of data, for practical reasons, we have opted for a not-so-abundant training data scenario. In typical wind resource assessment projects, one has access to merely one or two years of on-site data. The wind data analysts are then tasked

to build MCP models with such a limited amount of data. To mimic this situation, we train ML models with only one year of training data and, subsequently, make predictions for ten years. We repeat this process in a round-robin manner by changing the training and testing years. For example, in the schematic shown in Figure 3, we use data from the year 2003 for training and make predictions for the years 2004–2013.





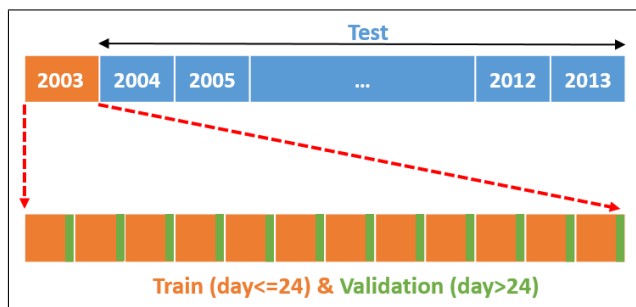

**Figure 3.** Our strategy of splitting the entire dataset into training, validation, and testing sets.

In ML training, it is customary to hold out a portion of the training data, called a validation set, for hyperparameter tuning.
Often an 80%-20% randomly shuffled split is made between training and validation sets. However, meteorological data are temporally correlated. Thus, random shuffling causes information leakage into the validation set. To minimize this undesirable leakage problem, we use the first twenty-four days of each month (i.e., $\sim 80\%$) for training and the rest for validation as depicted in Figure 3.

## 6.2   ML Models

In this study, we have used four different decision tree-based ML models. Two of them, Random Forest (Breiman, 2001) and Extremely Randomized Trees (Geurts et al., 2006), use the so-called bagging approach. The other two approaches, XGBoost (Chen and Guestrin, 2016; Wade, 2020) and LightGBM (Machado et al., 2019), are built on the gradient-boosting technique (Freund and Schapire, 1999; Friedman, 2002). For a comprehensive treatise on decision trees, bagging, and boosting, the following references are suggested: Rokach and Maimon (2008), Hastie et al. (2009), Géron (2022), and Murphy (2022). We
also encourage the readers to peruse the concise tutorial on decision-trees by Spiliotis (2022).

It is important to point out that we are interested in comparing the relative performance of various ML approaches for wind gust prediction and identifying if there is a clear winner. It is entirely possible that by combining some of these techniques (e.g., via a stacking regressor), one can get enhanced performance. However, we do not investigate this strategy in this paper.

## 6.3   Hyperparameter Tuning

Each DT-based model contains several hyperparameters (e.g., number of trees, number of tree levels). We include the most relevant ones in Table 2. Technical descriptions of these hyperparemeters are beyond the scope of this paper. The readers are encouraged to peruse the original publications and associated code repositories for more information.

In order to achieve high-level predictive performance, all the hyperparameters should be highly optimized. Quite often, random search or grid search approaches are used (Géron, 2022). These strategies are very time-consuming and may require
sophisticated hardware support. As an alternative, in this study, we have used FLAML (A Fast and Lightweight AutoML



Library), developed by Microsoft (https://microsoft.github.io/FLAML/). Instead of performing the grid search, the FLAML library takes the available computing time as a parameter and tries to find the optimal hyperparameters within the allotted time.

FLAML optimizes hyperparameters using effective search strategies. During the search process, the learner decides on the hyperparameter, sample size, and resampling strategy while taking advantage of the combined effects on both cost and error. 230 The design of the FLAML is presented in Figure 3 of Wang et al. (2021). It consists of two layers, an ML layer, and an AutoML layer. In the present study, since each ML model (i.e., RF, XGB) is optimized individually, the ML layer contains only the relevant model. The AutoML contains a learner proposer, a hyperparameter and sample size proposer, a resampling strategy proposer, and a controller. While the proposers are used to decide the variables, the controller is used to initiate the experiment using the learner selected in the ML layer. These steps are repeated during the allotted time. The algorithm uses the 235 random direct search method to decide hyperparameters (Wu et al., 2021).

In this study, we are focusing on three different WTM stations (REESE, MACY, and FLUVANNA). For each station, we have eleven distinct training sets (one for each year). For each training set, we have four DT-based candidate models. In summary, we have a total of $3 \times 11 \times 4 = 132$ cases of distinct hyperparameter optimizations. To limit the overall computing time, each case is optimized for one hour on a windows workstation equipped with an Intel Core i7 3.5 GHz CPU and NVIDIA GeForce 240 GTX 1070 (8 GB) GPU. The total computing time was 132 hours. As an example, we provide the best configuration values for the year 2003 in Table 2. In addition, we also provide the search range of each hyperparameter in this table.

### 6.4 Performance Evaluation Metrics

For model evaluations, we have used the mean absolute error (MAE), the mean squared error (MSE), and the coefficient of determination (R2) as metrics. They are defined as follows:

$$MAE = \frac{1}{N} \sum_{i=1}^{N} |y_i - \hat{y}| \tag{6a}$$

$$MSE = \frac{1}{N} \sum_{i=1}^{N} (y_i - \hat{y})^2 \tag{6b}$$

$$R^2 = 1 - \frac{\sum_{i=1}^{N} (y_i - \hat{y}_i)^2}{\sum_{i=1}^{N} (y_i - \bar{y})^2} \tag{6c}$$

250 where $y_i$ and $\hat{y}_i$, are the $i^{th}$ measured and the corresponding predicted values of $W_p$. The average of the measured $W_p$ values is denoted by $\bar{y}_i$. The total sample size in the test set is $N$. Since the overall test set consists of ten years of hourly data, $N$ is approximately equal to 8760 for each year.





## 7 Results

In this section, the predictive performances of four DT-based algorithms are evaluated for the three WTM stations (REESE, MACY, and FLUVANNA). In addition to these ML models, we use two ERA5 wind gust variables ($W_{p10}^i$, $W_{p10}^m$) as baseline predictors for $W_p$. Intuitively, we expect the ML models to outperform the ERA5 predictions as it uses more input features.

We first report the results for self-prediction cases where training and testing are performed using the WTM and ERA5 data from the same location. In the following sub-section, we discuss a cross-prediction scenario. Specifically, data from one of the WTM stations is used for training, and the fitted model is used to make predictions for the other two locations. In the last sub-section, we discuss the importance of various input features.

### 7.1 Self-Prediction

As mentioned earlier, one year of training data for each WTM station is used to build four DT-based models (optimized via FLAML). These site-specific tuned models are then used to predict $W_p$ for the other ten years for the same site. The mean prediction scores of all the models, in terms of MAE, MSE, and R2 metrics, are given in Tables 3, 4, and 5, respectively. As an illustrative example, let us consider the random forest (RF) model at REESE. Ten distinct RF models trained on data from the years 2004, 2005, ..., and 2013 are used for predicting $W_p$ for 2003. The average MAE from these ten predictions is 1.39 m s$^{-1}$.

From Tables 3 - 5, it is clear that the performance of ERA5's $W_{p10}^i$ and $W_{p10}^m$ variables as surrogates for $W_p$ exhibits an inter-annual variability. For example, for the $W_{p10}^i$ variable, the MAE at REESE ranges from 1.53 m s$^{-1}$–1.68$^{-1}$ with an average of 1.59 m s$^{-1}$. These tables also attest to the superior performance of the ERA5 baseline at REESE and FLUVANNA stations in comparison to MACY. Given the complex location of MACY and the coarse effective resolution of ERA5, such a deterioration in performance at MACY is expected.

All the performance metrics are considerably improved when using the DT based models instead of the ERA5 baseline. According to the numerical values given in Table 5, the XGB model improves the average R2 scores for REES, MACY, and FLUVANNA stations by 0.08, 0.11, and 0.11, respectively, in comparison to the ERA5-$W_{p10}^m$ baseline. The performances of the four DT-based ML models are pretty similar. In the case of REESE, the XGB model provides 12%, 13%, and 23% improvements in terms of R2, MAE, and MSE, respectively.

In Tables 3– 5, all the scores of the ML models are averaged over ten years. Thus, the inter-annual variability of all these models is much lower in comparison to the ERA5 baseline. For example, in the context of the XGB model, the R2 score at MACY has a narrow range of 0.68–0.70. To further investigate the year-to-year variability and performance of an ML model, we report the annual R2 scores at the MACY station in Table 6. As an illustrative example, we only tabulate the results of the XGB model. The results of the other ML models are very similar and, thus, are not shown. It is satisfying to see that the inter-annual variability of the R2 score is not more pronounced than the ERA5 baseline. In other words, with only one year of training data, the XGB model can estimate $W_p$ values for other years with R2 scores ranging from 0.63 to 0.73. These scores are considerably higher than the corresponding values (R2 = 0.52–0.61) from the ERA5-$W_{p10}^m$ baseline.





**Table 3.** MAE (m s$^{-1}$) scores of two baseline ERA5 variables and four DT-based models

| Station | Model | Years | | | | | | | | | | | |
|---|---|---|---|---|---|---|---|---|---|---|---|---|---|
| | | 2003 | 2004 | 2005 | 2006 | 2007 | 2008 | 2009 | 2010 | 2011 | 2012 | 2013 | Mean |
| REESE | $W_{p10}^i$ | 1.53 | 1.57 | 1.57 | 1.68 | 1.60 | 1.58 | 1.66 | 1.61 | 1.57 | 1.62 | 1.54 | 1.59 |
| | $W_{p10}^m$ | 1.49 | 1.54 | 1.55 | 1.65 | 1.61 | 1.61 | 1.67 | 1.62 | 1.58 | 1.60 | 1.54 | 1.59 |
| | RF | 1.39 | 1.45 | 1.39 | 1.40 | 1.40 | 1.40 | 1.47 | 1.41 | 1.39 | 1.38 | 1.39 | 1.41 |
| | ERT | 1.39 | 1.43 | 1.37 | 1.38 | 1.38 | 1.39 | 1.38 | 1.45 | 1.39 | 1.38 | 1.38 | 1.39 |
| | XGB | 1.39 | 1.46 | 1.41 | 1.37 | 1.37 | 1.39 | 1.36 | 1.39 | 1.41 | 1.37 | 1.38 | 1.39 |
| | LGBM | 1.43 | 1.45 | 1.39 | 1.38 | 1.40 | 1.45 | 1.37 | 1.41 | 1.39 | 1.37 | 1.38 | 1.40 |
| MACY | $W_{p10}^i$ | 1.89 | 1.93 | 1.84 | 1.92 | 1.89 | 1.97 | 1.94 | 1.82 | 1.83 | 1.91 | 1.78 | 1.88 |
| | $W_{p10}^m$ | 1.76 | 1.82 | 1.74 | 1.85 | 1.82 | 1.89 | 1.84 | 1.73 | 1.74 | 1.82 | 1.72 | 1.79 |
| | RF | 1.63 | 1.64 | 1.59 | 1.58 | 1.95 | 1.58 | 1.56 | 1.59 | 1.57 | 1.64 | 1.55 | 1.62 |
| | ERT | 1.60 | 1.61 | 1.56 | 1.55 | 1.55 | 1.57 | 1.56 | 1.57 | 1.57 | 1.55 | 1.56 | 1.57 |
| | XGB | 1.57 | 1.60 | 1.56 | 1.56 | 1.54 | 1.55 | 1.62 | 1.57 | 1.55 | 1.52 | 1.55 | 1.56 |
| | LGBM | 1.56 | 1.60 | 1.56 | 1.56 | 1.54 | 1.57 | 1.65 | 1.58 | 1.57 | 1.54 | 1.53 | 1.57 |
| FLUVANNA | $W_{p10}^i$ | 1.44 | 1.47 | 1.46 | 1.57 | 1.52 | 1.55 | 1.58 | 1.48 | 1.53 | 1.54 | 1.50 | 1.51 |
| | $W_{p10}^m$ | 1.47 | 1.51 | 1.51 | 1.65 | 1.56 | 1.62 | 1.62 | 1.51 | 1.56 | 1.56 | 1.51 | 1.55 |
| | RF | 1.31 | 1.34 | 1.31 | 1.27 | 1.29 | 1.28 | 1.28 | 1.29 | 1.33 | 1.32 | 1.32 | 1.30 |
| | ERT | 1.29 | 1.31 | 1.43 | 1.30 | 1.31 | 1.28 | 1.28 | 1.29 | 1.38 | 1.32 | 1.34 | 1.32 |
| | XGB | 1.29 | 1.28 | 1.31 | 1.27 | 1.28 | 1.26 | 1.28 | 1.28 | 1.34 | 1.30 | 1.34 | 1.29 |
| | LGBM | 1.29 | 1.29 | 1.32 | 1.31 | 1.30 | 1.30 | 1.28 | 1.29 | 1.33 | 1.28 | 1.35 | 1.30 |

## 7.2 Cross-Prediction

In order to demonstrate the potential generalizability of the ML models, the optimized models for the REESE station are utilized for predictions at MACY and FLUVANNA stations. The R2 scores are reported in Table 7. In the case self-prediction, the R2 scores for the ML models were around 0.66–0.69 for MACY and 0.72–0.73 for FLUVANNA (refer to Table 5). In the case of cross-prediction, the results are slightly poorer. In the case of MACY, the R2 values are approximately equal to 0.64; whereas, the corresponding R2 values are around 0.70–0.71 at FLUVANNA. These results are encouraging and imply that the proposed INTRIGUE approach might be used for cross-predictions as long as the training and testing locations are not too far apart and experience similar regional climatic conditions. Along this direction, more studies are needed for rigorous validations.

## 7.3 Feature Importance

In the INTRIGUE approach, we have used 265 input features. It is likely that not all these features are equally important for peak wind gust predictions. One way to rank the input features is via using the "permutation feature importance" strategy (Breiman, 2001; Molnar, 2022). To describe this simple algorithm, we closely follow Section 7.5 of Molnar (2022).





**Table 4.** MSE (m$^2$ s$^{-2}$) scores of two baseline ERA5 variables and four DT-based models

| Station | Model | Years | | | | | | | | | | | |
|---------|-------|------|------|------|------|------|------|------|------|------|------|------|------|
| | | 2003 | 2004 | 2005 | 2006 | 2007 | 2008 | 2009 | 2010 | 2011 | 2012 | 2013 | Mean |
| REESE | $W_{p10}^i$ | 4.51 | 4.93 | 4.85 | 5.78 | 5.15 | 4.92 | 5.42 | 5.13 | 4.52 | 4.97 | 4.69 | 4.99 |
| | $W_{p10}^m$ | 4.29 | 4.66 | 4.62 | 5.53 | 5.10 | 5.00 | 5.42 | 5.12 | 4.56 | 4.80 | 4.66 | 4.89 |
| | RF | 3.84 | 4.01 | 3.76 | 3.72 | 3.83 | 3.82 | 4.20 | 3.98 | 3.83 | 3.76 | 3.81 | 3.87 |
| | ERT | 3.80 | 3.92 | 3.72 | 3.66 | 3.79 | 3.81 | 3.75 | 4.26 | 3.82 | 3.78 | 3.75 | 3.82 |
| | XGB | 3.78 | 3.97 | 3.82 | 3.58 | 3.68 | 3.80 | 3.60 | 3.82 | 3.85 | 3.70 | 3.68 | 3.75 |
| | LGBM | 3.94 | 3.97 | 3.81 | 3.63 | 3.85 | 4.07 | 3.66 | 3.98 | 3.79 | 3.70 | 3.70 | 3.83 |
| MACY | $W_{p10}^i$ | 6.88 | 7.20 | 6.70 | 7.25 | 6.95 | 7.61 | 7.02 | 6.56 | 6.12 | 7.05 | 6.36 | 6.88 |
| | $W_{p10}^m$ | 6.13 | 6.51 | 6.08 | 6.70 | 6.52 | 7.10 | 6.46 | 5.97 | 5.63 | 6.43 | 5.93 | 6.31 |
| | RF | 4.93 | 5.04 | 4.76 | 4.74 | 6.71 | 4.69 | 4.66 | 4.87 | 4.81 | 5.16 | 4.67 | 5.00 |
| | ERT | 4.75 | 4.85 | 4.64 | 4.63 | 4.63 | 4.67 | 4.63 | 4.77 | 4.77 | 4.61 | 4.64 | 4.69 |
| | XGB | 4.60 | 4.76 | 4.59 | 4.55 | 4.53 | 4.60 | 4.89 | 4.75 | 4.61 | 4.51 | 4.60 | 4.64 |
| | LGBM | 4.56 | 4.77 | 4.59 | 4.57 | 4.58 | 4.66 | 5.09 | 4.76 | 4.74 | 4.59 | 4.54 | 4.68 |
| FLUVANNA | $W_{p10}^i$ | 4.06 | 4.14 | 4.38 | 4.85 | 4.71 | 4.77 | 4.78 | 4.34 | 4.38 | 4.71 | 4.60 | 4.52 |
| | $W_{p10}^m$ | 4.04 | 4.27 | 4.47 | 5.14 | 4.87 | 5.02 | 4.96 | 4.36 | 4.55 | 4.71 | 4.55 | 4.63 |
| | RF | 3.35 | 3.54 | 3.35 | 3.22 | 3.28 | 3.21 | 3.26 | 3.26 | 3.37 | 3.39 | 3.30 | 3.32 |
| | ERT | 3.28 | 3.40 | 4.04 | 3.35 | 3.40 | 3.25 | 3.26 | 3.28 | 3.55 | 3.35 | 3.37 | 3.41 |
| | XGB | 3.25 | 3.21 | 3.31 | 3.20 | 3.16 | 3.15 | 3.22 | 3.17 | 3.35 | 3.23 | 3.35 | 3.24 |
| | LGBM | 3.29 | 3.27 | 3.39 | 3.36 | 3.31 | 3.30 | 3.22 | 3.23 | 3.36 | 3.22 | 3.40 | 3.30 |

First, an ML model (say XGB) is trained using one year of data from a specific station (e.g., REESE). Then, we make a
prediction for another year for the same station. Both the training and testing data contain 265 input features. Using the observed
and predicted $W_p$ values, we compute prediction errors (e.g., using R2) and denote this error as $e_o$. Next, we randomly shuffle
only one of the input features (say the $i$-th feature) of the test data and keep the ordering of all other features the same. Now,
we make a new prediction. The error associated with this new prediction is denoted as $e_p^i$. Since we have randomized only one
input feature, that feature no longer has any association with the other input features. Thus, we expect $e_p^i$ to be worse than $e_o$;
in the case of R2, $e_p^i \leq e_o$. To achieve converged statistics, we repeat the randomization process for the same $i$-th feature a few
times (typically 5 or more) and compute an averaged value of $\overline{e_p^i}$. The net reduction in R2 score due to the randomization of
$i$-th feature is: $\left( e_o - \overline{e_p^i} \right)$.

One at a time, we repeat the random shuffling exercise for all 265 input features and compute the reduction in R2 corre-
sponding to each input feature. If an input feature is very important for peak wind gust estimation, the reduction in R2 for that
feature will be large. On the other hand, the irrelevant input features marginally impact the R2 scores.

In Figure 4, the importance (in terms of reduction in R2) of all the input features is plotted for the XGB and RF models. For
computation, we use the ELI5 library (https://eli5.readthedocs.io/en/latest/overview.html). We average the statistics over ten
years for robustness.



**Table 5.** R2 scores of two baseline ERA5 variables and four DT-based models

| Station | Model | Years | | | | | | | | | | | |
|---|---|---|---|---|---|---|---|---|---|---|---|---|---|
| | | 2003 | 2004 | 2005 | 2006 | 2007 | 2008 | 2009 | 2010 | 2011 | 2012 | 2013 | Mean |
| REESE | $W_{p10}^i$ | 0.69 | 0.61 | 0.64 | 0.60 | 0.60 | 0.70 | 0.62 | 0.65 | 0.70 | 0.66 | 0.67 | 0.65 |
| | $W_{p10}^m$ | 0.70 | 0.63 | 0.65 | 0.62 | 0.61 | 0.69 | 0.62 | 0.65 | 0.70 | 0.67 | 0.67 | 0.66 |
| | RF | 0.73 | 0.72 | 0.74 | 0.74 | 0.73 | 0.73 | 0.71 | 0.72 | 0.73 | 0.73 | 0.73 | 0.73 |
| | ERT | 0.73 | 0.73 | 0.74 | 0.74 | 0.74 | 0.73 | 0.74 | 0.70 | 0.73 | 0.73 | 0.74 | 0.73 |
| | XGB | 0.73 | 0.72 | 0.73 | 0.75 | 0.74 | 0.73 | 0.75 | 0.73 | 0.73 | 0.74 | 0.74 | 0.74 |
| | LGBM | 0.72 | 0.72 | 0.73 | 0.74 | 0.73 | 0.71 | 0.74 | 0.72 | 0.73 | 0.74 | 0.74 | 0.73 |
| MACY | $W_{p10}^i$ | 0.52 | 0.47 | 0.52 | 0.50 | 0.52 | 0.58 | 0.54 | 0.55 | 0.62 | 0.53 | 0.57 | 0.54 |
| | $W_{p10}^m$ | 0.58 | 0.52 | 0.56 | 0.54 | 0.55 | 0.61 | 0.58 | 0.59 | 0.65 | 0.57 | 0.60 | 0.58 |
| | RF | 0.67 | 0.67 | 0.68 | 0.68 | 0.55 | 0.68 | 0.69 | 0.67 | 0.68 | 0.65 | 0.69 | 0.66 |
| | ERT | 0.68 | 0.68 | 0.69 | 0.69 | 0.69 | 0.68 | 0.69 | 0.68 | 0.68 | 0.69 | 0.69 | 0.69 |
| | XGB | 0.69 | 0.68 | 0.69 | 0.70 | 0.70 | 0.69 | 0.67 | 0.68 | 0.69 | 0.70 | 0.69 | 0.69 |
| | LGBM | 0.70 | 0.68 | 0.69 | 0.69 | 0.69 | 0.68 | 0.66 | 0.68 | 0.68 | 0.69 | 0.70 | 0.69 |
| FLUVANNA | $W_{p10}^i$ | 0.65 | 0.61 | 0.60 | 0.58 | 0.60 | 0.66 | 0.62 | 0.64 | 0.67 | 0.61 | 0.64 | 0.63 |
| | $W_{p10}^m$ | 0.65 | 0.60 | 0.60 | 0.55 | 0.59 | 0.64 | 0.60 | 0.64 | 0.66 | 0.61 | 0.64 | 0.62 |
| | RF | 0.72 | 0.71 | 0.73 | 0.73 | 0.73 | 0.73 | 0.73 | 0.73 | 0.72 | 0.72 | 0.73 | 0.73 |
| | ERT | 0.73 | 0.72 | 0.67 | 0.72 | 0.72 | 0.73 | 0.73 | 0.73 | 0.70 | 0.72 | 0.72 | 0.72 |
| | XGB | 0.73 | 0.74 | 0.73 | 0.74 | 0.74 | 0.74 | 0.73 | 0.74 | 0.72 | 0.73 | 0.72 | 0.73 |
| | LGBM | 0.73 | 0.73 | 0.72 | 0.72 | 0.73 | 0.72 | 0.73 | 0.73 | 0.72 | 0.73 | 0.72 | 0.73 |

**Table 6.** Detailed R2 scores of the XGB model at the MACY station for each year

| Training Years | Testing Years | | | | | | | | | | |
|---|---|---|---|---|---|---|---|---|---|---|---|
| | 2003 | 2004 | 2005 | 2006 | 2007 | 2008 | 2009 | 2010 | 2011 | 2012 | 2013 |
| $W_{p10}^i$ | 0.52 | 0.47 | 0.52 | 0.50 | 0.52 | 0.58 | 0.54 | 0.55 | 0.62 | 0.53 | 0.57 |
| $W_{p10}^m$ | 0.58 | 0.52 | 0.56 | 0.54 | 0.55 | 0.61 | 0.58 | 0.59 | 0.65 | 0.57 | 0.60 |
| 2003 | - | 0.67 | 0.69 | 0.67 | 0.67 | 0.74 | 0.69 | 0.69 | 0.74 | 0.68 | 0.70 |
| 2004 | 0.70 | - | 0.68 | 0.65 | 0.66 | 0.73 | 0.67 | 0.68 | 0.71 | 0.67 | 0.70 |
| 2005 | 0.70 | 0.67 | - | 0.66 | 0.66 | 0.73 | 0.69 | 0.68 | 0.74 | 0.69 | 0.71 |
| 2006 | 0.71 | 0.66 | 0.69 | - | 0.66 | 0.74 | 0.69 | 0.68 | 0.74 | 0.69 | 0.71 |
| 2007 | 0.71 | 0.66 | 0.69 | 0.66 | - | 0.73 | 0.70 | 0.69 | 0.74 | 0.69 | 0.70 |
| 2008 | 0.69 | 0.65 | 0.68 | 0.66 | 0.66 | - | 0.68 | 0.70 | 0.73 | 0.68 | 0.71 |
| 2009 | 0.69 | 0.63 | 0.68 | 0.63 | 0.65 | 0.72 | - | 0.67 | 0.69 | 0.66 | 0.70 |
| 2010 | 0.68 | 0.64 | 0.67 | 0.65 | 0.65 | 0.72 | 0.68 | - | 0.74 | 0.68 | 0.70 |
| 2011 | 0.69 | 0.66 | 0.68 | 0.67 | 0.66 | 0.73 | 0.70 | 0.70 | - | 0.69 | 0.71 |
| 2012 | 0.70 | 0.66 | 0.70 | 0.67 | 0.66 | 0.73 | 0.70 | 0.71 | 0.74 | - | 0.71 |
| 2013 | 0.69 | 0.66 | 0.69 | 0.66 | 0.67 | 0.74 | 0.69 | 0.70 | 0.74 | 0.68 | - |





**Table 7.** R2 scores of the two baseline ERA5 variables and four DT-based models at MACY and FLUVANNA. Models are trained using data from the REESE station.

| Station | Model | Years | | | | | | | | | | | |
|---|---|---|---|---|---|---|---|---|---|---|---|---|---|
| | | 2003 | 2004 | 2005 | 2006 | 2007 | 2008 | 2009 | 2010 | 2011 | 2012 | 2013 | Mean |
| MACY | $W_{p10}^i$ | 0.52 | 0.47 | 0.52 | 0.50 | 0.52 | 0.58 | 0.54 | 0.55 | 0.62 | 0.53 | 0.57 | 0.54 |
| | $W_{p10}^m$ | 0.58 | 0.52 | 0.56 | 0.54 | 0.55 | 0.61 | 0.58 | 0.59 | 0.65 | 0.57 | 0.60 | 0.58 |
| | RF | 0.64 | 0.64 | 0.64 | 0.63 | 0.65 | 0.63 | 0.62 | 0.64 | 0.64 | 0.64 | 0.65 | 0.64 |
| | ERT | 0.63 | 0.65 | 0.65 | 0.63 | 0.66 | 0.63 | 0.65 | 0.62 | 0.63 | 0.65 | 0.65 | 0.64 |
| | XGB | 0.64 | 0.65 | 0.64 | 0.66 | 0.64 | 0.64 | 0.66 | 0.65 | 0.64 | 0.66 | 0.66 | 0.65 |
| | LGBM | 0.62 | 0.64 | 0.65 | 0.65 | 0.62 | 0.62 | 0.65 | 0.64 | 0.63 | 0.64 | 0.65 | 0.64 |
| FLUVANNA | $W_{p10}^i$ | 0.65 | 0.61 | 0.60 | 0.58 | 0.60 | 0.66 | 0.62 | 0.64 | 0.67 | 0.61 | 0.64 | 0.63 |
| | $W_{p10}^m$ | 0.65 | 0.60 | 0.60 | 0.55 | 0.59 | 0.64 | 0.60 | 0.64 | 0.66 | 0.61 | 0.64 | 0.62 |
| | RF | 0.71 | 0.70 | 0.71 | 0.69 | 0.71 | 0.70 | 0.69 | 0.70 | 0.71 | 0.71 | 0.71 | 0.70 |
| | ERT | 0.70 | 0.71 | 0.71 | 0.69 | 0.72 | 0.70 | 0.71 | 0.68 | 0.71 | 0.71 | 0.72 | 0.70 |
| | XGB | 0.71 | 0.70 | 0.71 | 0.72 | 0.70 | 0.71 | 0.72 | 0.71 | 0.71 | 0.70 | 0.72 | 0.71 |
| | LGBM | 0.68 | 0.70 | 0.70 | 0.71 | 0.66 | 0.68 | 0.71 | 0.70 | 0.71 | 0.69 | 0.71 | 0.70 |

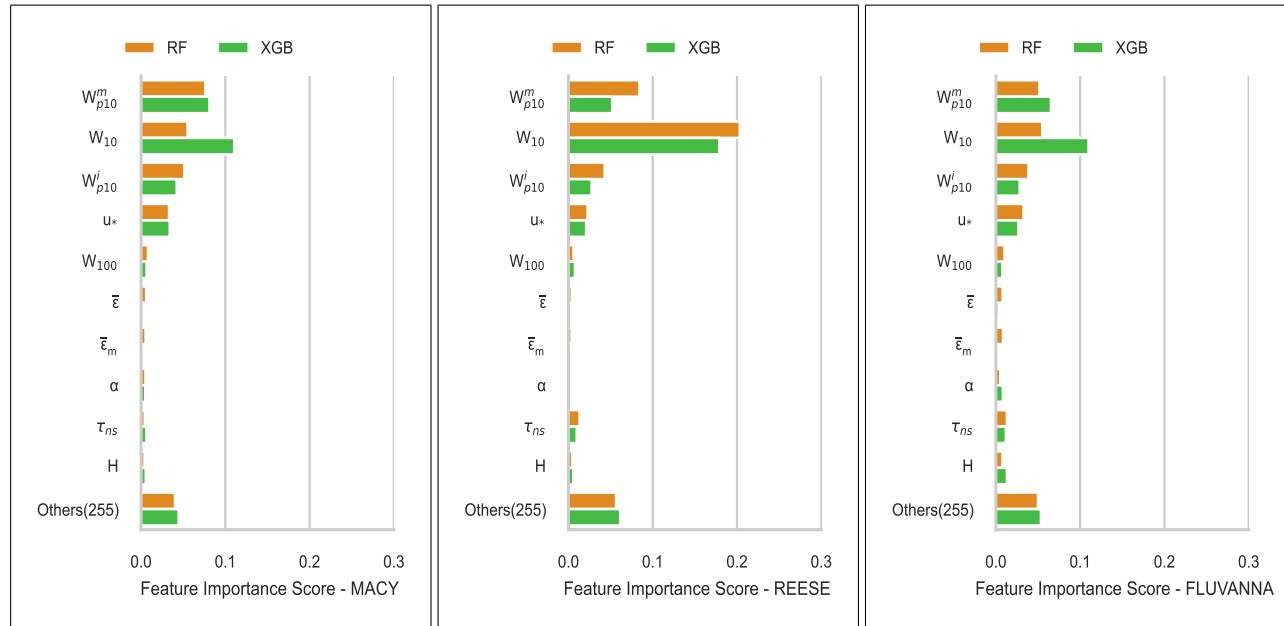

**Figure 4.** Feature importance scores of the ERA5 parameters for REESE (left panel), MACY (middle panel), and FLUVANNA (right panel). The results from two ML models, XGB and RF, are shown for comparison.

Although there are differences in the magnitude of the feature importance depending on the stations, the following input features are found to be very relevant for all three stations: $W_{10}, W_{p10}^m, W_{p10}^i, u_*, \tau_{ns}, W_{100}, \overline{\varepsilon}, \overline{\varepsilon}_m$, and $\alpha$. Interestingly, both



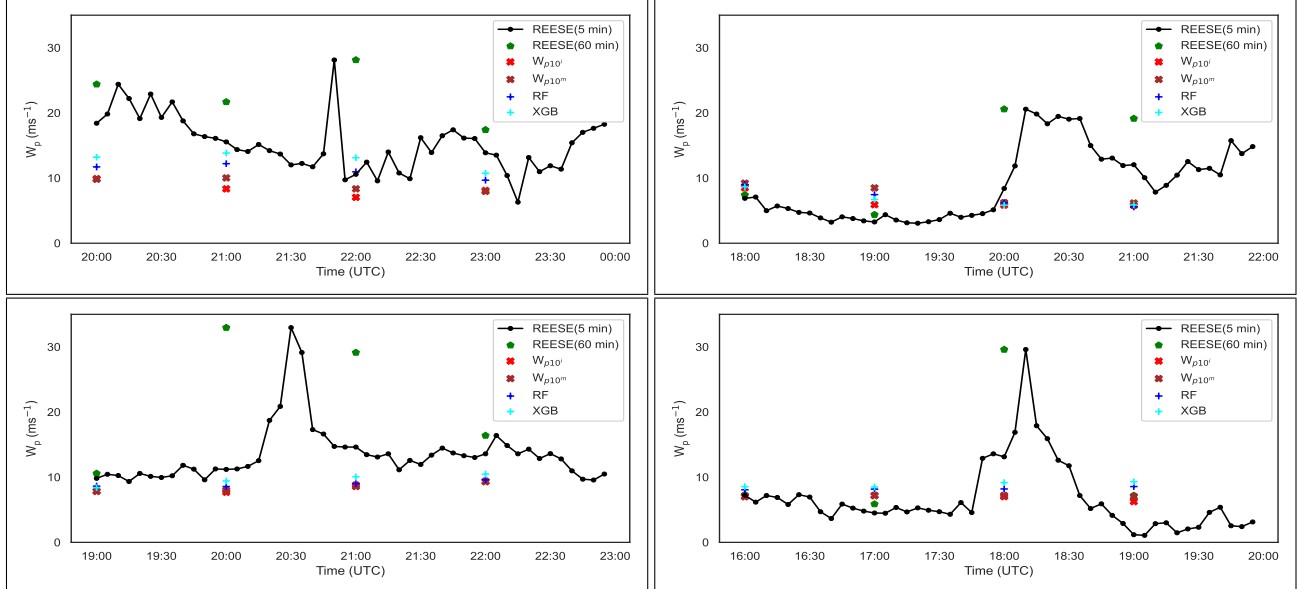

**Figure 5.** Examples of extreme wind gust events measured at the REESE station on June 19, 2008 (top-left panel; non-supercell thunderstorm), August 14, 2008 (top-right panel; non-supercell thunderstorm), June 4, 2009 (bottom-left panel; bow-echo/supercell thunderstorm), and August 12, 2009 (bottom-right panel; non-supercell thunderstorm). On these figures, the instantaneous wind gust ($W_{p10}^i$) values from the ERA5 dataset are overlayed for comparison. In addition, we have plotted the predictions from the RF and XGB models.

the XGB and RF models capture the same behavior. These input features are also the ones that are commonly used in physical parameterizations (see Section 2).

Some of the input features (e.g., related to the time of day, temperature, cloud cover) are not relevant for peak wind gust predictions. Thus, one can remove these input features from future ML models and achieve a similar level of prediction accuracy with reduced computational costs.

# 8   Limitations of the INTRIGUE Approach

The WTM dataset contains a handful of extreme wind gust events. In Figure 5, a few illustrative cases measured at the REESE station are shown. One of these cases is related to a supercell thunderstorm, while the others are produced by non-supercell thunderstorm events. These cases and a few others were studied in-depth by Lombardo et al. (2014). On these plots, we have overlaid $W_{p10}^i$ values from the ERA5 and also the predictions from two of the ML models (i.e., RF and XGB). It is apparent that the $W_{p10}^i$ variable has not captured the extreme wind gusts in a faithful manner. This failure is likely due to the coarse effective resolution of the ERA5 data, which cannot resolve thunderstorms. The ML models are unable to make any improvement to these extreme wind gust predictions.





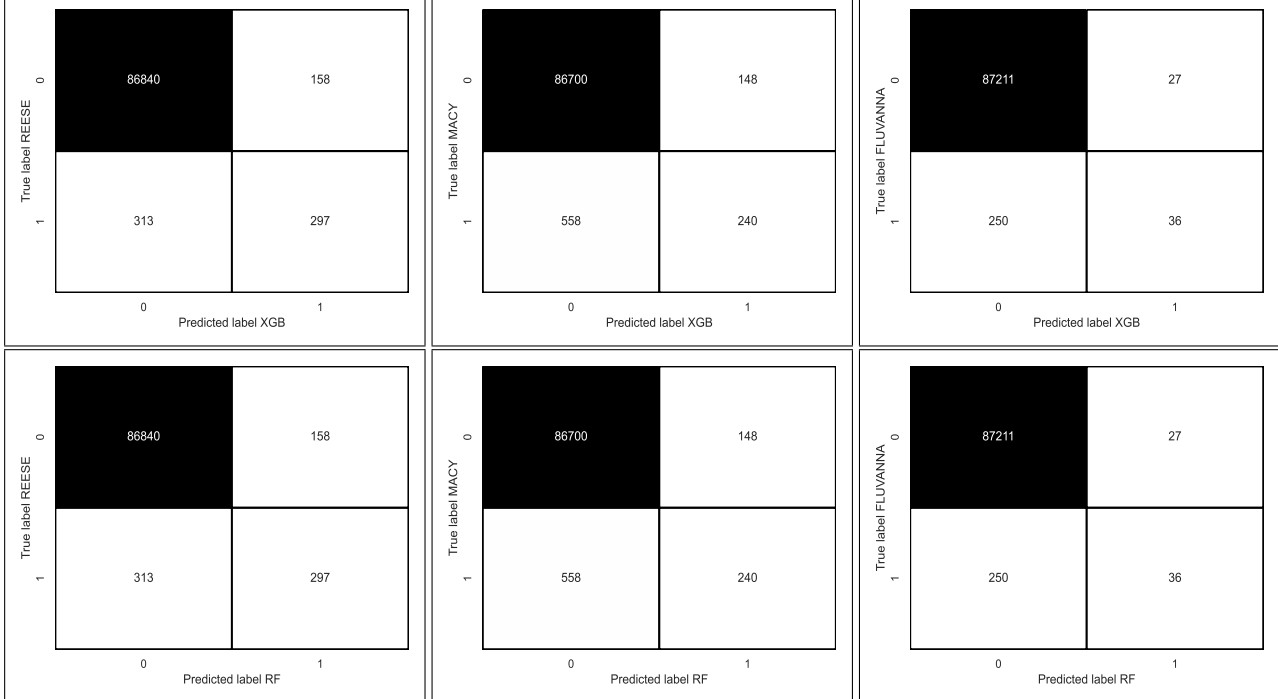

**Figure 6.** Confusion matrices for extreme wind gust ($W_p > 20$ m s$^{-1}$) prediction. The top and bottom panels represent XGBoost and RF models, respectively. The left, middle, and right panels correspond to REESE, MACY, and FLUVANNA stations, respectively.

To further investigate this limitation of the INTRIGUE approach, we provided several confusion matrices in Figure 6. We classified peak wind gusts into extreme (1) and nominal (0). When $W_p$ exceeds 20 m s$^{-1}$, we denote the event as an extreme. From these matrices, it is evident that the INTRIGUE approach leads to numerous false positives and false negatives.

In Figure 7, we show scatter plots of a few input features (or predictors) and the predictand ($W_p$). While discussing feature importance, we demonstrated that overall $W_{10}$, $W_{p10}^i$, and $W_{p10}^m$, are very important features. However, these features are barely correlated with $W_p$ for extreme conditions. Furthermore, ERA5's CAPE variable (typically related to thunderstorm development) is also not well-correlated with $W_p$ values. In lieu of adequate input features, the INTRIGUE approach fails to perform satisfactorily for the extreme wind gust conditions.

## 9 Conclusions

In this study, we proposed a decision tree-based MCP approach (called INTRIGUE) for peak wind gust estimation. This approach utilizes several meteorological variables (including the instantaneous wind gust variable) from the ERA5 reanalysis dataset as input features. For non-extreme (i.e., nominal) cases, the INTRIGUE approach-predicted peak wind gust values are closer to the observed ones than the baseline approaches. This approach can also make predictions for neighboring stations where training data is not available.

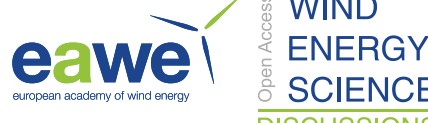

**Figure 7.** Scatter plots of $W_{10}$ (top panels), $W_{p10}^i$ (middle panels), and CAPE (bottom panels) against $W_p$ measured at REESE station. The $W_p$ values greater than 20 m s$^{-1}$ are only included in these plots. The left, middle, and the right panels correspond to years 2003, 2008, and 2013, respectively. There are only 77, 78, and 63 samples in the left, middle, and the right panels, respectively. It is clear that the correlations between the predictors and the predictand are very low for all the cases.

However, there is room for significant improvements as the INTRIGUE approach drastically underestimates extreme wind gust events of magnitudes higher than 20 m s$^{-1}$. For these cases, none of the 265 input features, that we considered in this
study, correlate with $W_p$. Clearly, we need more relevant input features. In our future work, we will also analyze meteorological profiles from ERA5 and compute various thunderstorm-related parameters as input features. In addition, we will add input features extracted from radar reflectivity fields using autoencoders. We speculate that the addition of such input features will enable the INTRIGUE approach to capture extreme wind gusts in a more faithful manner.

We would like to remind the readers that we intentionally use only one year of training data in this study. As a result, only a
few such extreme cases (on the order of 60-80 samples) are included in the training process. In the ML literature, this problem is known as the imbalance data problem. In the future, we will explore various ML strategies (e.g., isolation forest) to tackle this challenging problem.

In typical wind energy projects, one does not have access to on-site long-term wind gust datasets. Thus, increasing the sample size from a single site is not a viable solution. However, it will be possible to increase the sample size by aggregating
observational data from different sites across the world. By doing so, we will be able to come up with a more generalized ML model for wind gust prediction. We will pursue this line of research in the near future.

*Code availability.* Code will be available from github after publication.

*Data availability.* The ERA5 reanalysis data are provided by the European Centre for Medium- Range Weather Forecasts (https://cds.climate.copernicus.e

*Data availability.* The mesonet datasets are available from West Texas Mesonet (www.mesonet.ttu.edu).

*Author contributions.* SK and SB were responsible for the overall conceptualization of the study. SK wrote all the computer codes and performed all the data analysis. SK, SB, and SJW were involved in the writing, and editing of the manuscript

*Competing interests.* Some authors are members of the editorial board of Wind Energy Science. The peer-review process was guided by an independent editor, and the authors have also no other competing interests to declare.

*Acknowledgements.* The authors are grateful to West Texas Mesonet for sharing their datasets. This research was partially supported by the
TU Delft Institute for Computational Science and Engineering (DCSE).



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
