# Peer review of "A decision tree-based measure-correlate-predict approach for peak wind gust estimation from a global reanalysis dataset"

_Wind Energy Science, 2023_

## Referee Comment (RC1)

I am happy to recommend the manuscript for publication in Wind Energy Science, as the subject and the approach used in the paper will be of broad international interest. I also strongly encourage publishing the code on github. My main remarks are listed below.

**Specific comments:**

- **Section 7.1 Self-Prediction:** Although the main ideas have been introduced, the overall presentation of this section still needs improvement. In particular, please clarify which data are used as training data and which are used for prediction. In particular, reconcile this section with Section 6.1 "Strategy for Splitting of Available Data".

- **Table 3-5, 7:** Highlight the essential items in the tables, such as the higher correlation of the means, or choose another more illustrative form of presentation.

- **Figure 6:** In addition to or instead of the confusion matrix, present the respective receiver operating characteristic curve.

- **Section 8 Limitations of the INTRIGUE Approach:** Add a discussion about which input features would most likely enable the INTRIGUE approach to predict wind gusts for extreme conditions.

**Technical comments:**

- **Line 3:** Define short & long-term in the manuscript.

- **Line 3:** Define nominal & extreme conditions in the manuscript.

- **Line 27:** The sentence: "The focus of the current... approach." might be better linked to the sentence: "In this paper, we propose a... reanlysis dataset." (Line 79).

- **Line 55 - 69:** This section could better follow directly after the section that ends at line 27.

- **Table 1:** Highlight most important input features for the approach

- **Figure 2, 7:** Add a liner relationship to each of the graphs to illustrate the correlation.

- **Figure 5:** Add Definition of $W_{p10^m}$ in the caption

- **Figure 6:** Are the Confusion Matrix actually the exact same for the XG-Boost and RF models?

- **Line 355:** Add the note that for the cross-predictions the different sites across the world should experience similar regional conditions.

---

## Author Comment (AC1)

**Response to Reviewer 1**

We thank the reviewer for their time and the evaluations of our paper. We have carefully read the comments (shown in blue font) and provide point-by-point response here (in black font).

General comments:
The manuscript titled "A decision tree-based measure-correlate-predict approach for peak wind gust estimation from a global reanalysis dataset" by S. Kartal et al. submitted for publication in Wind Energy Science adopts a new decision tree (DT)-based Measure-Correlate-Predict (MCP) approach called INTRIGUE that utilizes several meteorological variables from a public-domain reanalysis dataset to generate long-term, site-specific peak wind gust (Wp) estimation. The intro duced four different models perform very well for nominal (non-extreme) conditions but the full potential for extreme conditions is yet to be seen. In particular, the application of this approach in the context of wind energy is very interesting, as wind gusts are critical for the highly fluctuating power production. The work presented here may serve as yet another approach to address this open issue.

We thank the reviewer for the summary and encouraging remarks.

I am happy to recommend the manuscript for publication in Wind Energy Science, as the subject and the approach used in the paper will be of broad international interest. I also strongly encourage publishing the code on github. My main remarks are listed below.

We thank the reviewer for his positive recommendation. We have already published the codes at:
`https://github.com/serkankartal/PeakWindGustEstimation`.

Specific comments:

Section 7.1 Self-Prediction: Although the main ideas have been introduced, the overall presentation of this section still needs improvement. In particular, please clarify which data are used as training data and which are used for prediction. In particular, reconcile this section with Section 6.1 "Strategy for Splitting of Available Data".

We agree that the original text was a bit confusing. We have expanded on the description in the revised manuscript (see the text marked in blue).

Table 3-5, 7: Highlight the essential items in the tables, such as the higher correlation of the means, or choose another more illustrative form of presentation.

Following the suggestions of both the reviewers, we have replaced the tables by bar plots in the revised manuscript.

Figure 6: In addition to or instead of the confusion matrix, present the respective receiver operating characteristic curve.

The ROC curves are useful when one has multiple thresholds in a binary classification problem. In this work, we only use one threshold value of 20 m s$^{-1}$ to distinguish between nominal and extreme wind speeds. Hence, we decided not to add any ROC curves.

Section 8 Limitations of the INTRIGUE Approach: Add a discussion about which input features would most likely enable the INTRIGUE approach to predict wind gusts for extreme conditions.

We added the following sentence:

> "We speculate that parameters derived from vertical profiles of the ERA5 reanalysis (e.g., deep-layer wind shear, storm relative helicity, integrated Scorer parameter, Sangster parameter) as input features might improve the predictions."

Technical comments:
Line 3: Define short & long-term in the manuscript.

We replaced 'long-term' with 'multi-year'.

Line 3: Define nominal & extreme conditions in the manuscript.

We have rephrased this sentence as follows:

> "The INTRIGUE approach outperforms the baseline predictions for all wind gust conditions. However, the performance of this proposed approach and the baselines for extreme conditions (i.e., $W_p > 20$ m s$^{-1}$) is less than satisfactory."

Line 27: The sentence: "The focus of the current... approach." might be better linked to the sentence: "In this paper, we propose a... reanlysis dataset." (Line 79).

We decided not to make this change. The line 27 briefly introduces the reader to the overall focus. Then, after we introduce various technical details (e.g.,

Eqs. 1 and 2), we give a more technical description.

Line 55 - 69: This section could better follow directly after the section that ends at line 27.

Moving this paragraph after line 27 will likely disrupt the flow. We want to describe the importance of nominal (lines 48-54) and extreme (lines 55-69) wind gusts following one another. For this reason, we did not revise the text.

Table 1: Highlight most important input features for the approach.

We have printed the important variables in bold. These variables were identified via permutation feature importance analysis.

Figure 2, 7: Add a liner relationship to each of the graphs to illustrate the correlation.

Following the reviewer's suggestion, we have included linear regression fits in these plots.

Figure 5: Add Definition of Wp10m in the caption.

We have added the definition.

Figure 6: Are the Confusion Matrix actually the exact same for the XG Boost and RF models?

We thank the reviewer for catching this plotting error. We have revised the plots.

Line 355: Add the note that for the cross-predictions the different sites across the world should experience similar regional conditions.

We added 'comparable climatic conditions' in the revised manuscript.

---

## Author Comment (AC2)

**Response to Reviewer 2**

*This paper introduces a decision tree-based application of the measure-correlate-predict methodology, which is commonly employed in the wind industry, to estimate wind gusts. The manuscript is very well written and provides interesting historical context. The machine learning approaches outperform ERA5 in wind gust representation at three observational locations in West Texas and predictor variables are ranked in order of importance to the algorithms.*

We thank the reviewer for the succinct summary and positive remarks.

*The manuscript, while quite interesting, would benefit from some discussion about the potential applicability of these gust estimation techniques to the wind energy audience of this journal. To elaborate, the tests performed in this work are at 10 m above ground level, while typical turbine hub heights are 80+ m above ground level. Do the authors have any speculation or insight into how the performance of their models and the importance scores of the various parameters might change for a higher height?*

We have elaborated on this issue in Section 8 of the revised manuscript.

*Additionally, it would be helpful to understand the ultimate goal of this research to support the wind community. Is the aim to improve gust estimates in wind farm forecasts? Or to provide long-term assessment of the frequency and magnitude of gusts at a proposed wind farm? Lines 198-201 hint at the latter, however, an explicit goal statement would be advantageous to the text.*

In the abstract, we clearly mentioned that our goal is to generate long-term, site-specific wind gusts data via machine learning. Towards the end of Section 3, we also mentioned that we are proposing a measure-correlate-predict (MCP) approach for wind gusts. See also the first sentence in Conclusions.

In addition to site assessments, our proposed INTRIGUE approach can be used in wind gust forecasting. Instead of a reanalysis dataset, predicted meteorological fields from a numerical weather prediction model can be used as input features for the ML models. We added this information in the revised manuscript.

*My one concern with the analysis is the method selected for comparison of time-series of varying temporal resolution (Lines 182-187). Taking the maximum gust from the 5-minute observations within each hour and assigning it to the top of the hour for evaluation of hourly models seems an unfair comparison, which is particularly noticeable in Figure 5. Was there a reason you did not choose the observed wind gust closest to the top of the hour for your comparisons?*

The reviewer raises an important issue. Unfortunately, there is no fool-proof approach in the literature for comparing high-frequency point observations and

numerically modeled data that are spatially filtered. Our strategy had two justifications.

First, the proposed decision tree-based approaches utilize the variable $W_{p10}^m$ (called $10fg$ in ERA5) as one of the input features. This particular variable represents mean wind gusts during the past hour and, thus, is expected to be correlated with the observed maximum gusts during the same period.

Second, the observed wind gusts are spatiotemporally intermittent. Therefore, we were concerned that by simple temporal downsampling, we would lose a significant number of extreme gust events. Averaging the observed 5-min gusts to the hourly averaged value was not considered a viable alternative, as this approach would have drastically reduced the amplitude of extreme wind gusts. Thus, we opted for taking the maximum value over the past hour. We would like to point out that each grid point of ERA5 effectively represents a spatial coverage of approximately 32 km x 32 km; hence, wind gusts from the ERA5 reanalysis are expected to be much less intermittent than observations.

Other comments: Line 102: "Quasi-universal" is a bit of a strong statement given the small sample size.

We changed it to 'not site-specific'.

Line 177: The authors should consider elaborating on why ERA5 was selected, including citing wind studies that employ it. Additionally, I think a literature review discussion on the accuracy of some of the ERA5 variables employed as predictors in the analysis, in particular the ERA5 instantaneous wind gust, friction velocity, and boundary layer height.

Soon after its introduction, the ERA5 dataset became the preferred reanalysis dataset in the wind power meteorology community. The following papers (and many others) discuss its superior accuracy, lower uncertainty, and higher reliability compared to other global reanalysis datasets.

1. Olauson, J. (2018). ERA5: The new champion of wind power modelling?. Renewable Energy, 126, 322-331.

2. Ramon, J., Lledó, L., Torralba, V., Soret, A., & Doblas-Reyes, F. J. (2019). What global reanalysis best represents near-surface winds? Quarterly Journal of the Royal Meteorological Society, 145, 3236-3251.

3. Gualtieri, G. (2022). Analysing the uncertainties of reanalysis data used for wind resource assessment: A critical review. Renewable and Sustainable Energy Reviews, 167, 112741.

We have included these references in the revised manuscript. It is beyond the scope of this paper to discuss the accuracy of the ERA5 variables.

Line 180: What is the distance between each observation station and its nearest ERA5 point?

The distances between the REESE, MACY, FLUVANNA stations and their corresponding ERA5 grid points are 14 km, 9 km, and 12 km, respectively. We have added this information in the revised manuscript.

Section 6.4: I encourage adding bias to the list of performance metrics, as it would be valuable for the wind community to understand whether the evaluated models tend to overestimate or underestimate observed gusts.

We have added bias as an additional performance metrics.

Line 268/Tables 3-5 (and 7): "From Tables 3-5, it is clear. . . " would be clearer if these were figures instead of tables. For example, each one could be a line plot with subplots a, b, c for the different stations.

Following the suggestions of both reviewers, we have replaced the tables with bar diagrams in the revised manuscript.

Line 278: "In Tables 3– 5, all the scores of the ML models are averaged over ten years. Thus, the inter-annual variability of all these models is much lower in comparison to the ERA5 baseline." These sentences are confusing. Is the inter-annual variability is lower because all of the scores are averaged or because of the model performance?

We agree that the original sentences were a bit confusing. We have rewritten them in the revised manuscript as follows:

> "In Figures 5–7, all the scores of the ML models are averaged over ten years. Due to averaging, the perceived inter-annual variability of all these models is much lower in comparison to the ERA5 baseline. For example, in the context of the XGB model, the R2 score at MACY has a narrow range of 0.68–0.70. In order to investigate the year-to-year variability and performance of an ML model, we report the annual R2 scores at the MACY station in Table 3."

Section 7.2: This section is quite interesting, given the challenges of observational coverage. I encourage investigation of geographic range of applicability of the discussed techniques for a future paper.

We do intend to expand our work in this direction.